# Machine Learning Algorithms: Prediction and Feature Selection for Clinical Refracture after Surgically Treated Fragility Fracture

**DOI:** 10.3390/jcm11072021

**Published:** 2022-04-05

**Authors:** Hirokazu Shimizu, Ken Enda, Tomohiro Shimizu, Yusuke Ishida, Hotaka Ishizu, Koki Ise, Shinya Tanaka, Norimasa Iwasaki

**Affiliations:** 1Department of Orthopaedic Surgery, Faculty of Medicine and Graduate School of Medicine, Hokkaido University, Sapporo 060-8638, Japan; rottefella.nnn@gmail.com (H.S.); iszhtk24@gmail.com (H.I.); niwasaki@med.hokudai.ac.jp (N.I.); 2Department of Cancer Pathology, Faculty of Medicine, Hokkaido University, Sapporo 060-8638, Japan; buhibuhidog@gmail.com (K.E.); isek0k126@gmail.com (K.I.); tanaka@med.hokudai.ac.jp (S.T.); 3Department of Pathology, Hokkaido Medical Center, Sapporo 063-0005, Japan; yuishida@abox9.so-net.ne.jp

**Keywords:** machine learning algorithms, fragility fracture, clinical refracture, LightGBM, rheumatoid arthritis, chronic kidney disease

## Abstract

Background: The number of patients with fragility fracture has been increasing. Although the increasing number of patients with fragility fracture increased the rate of fracture (refracture), the causes of refracture are multifactorial, and its predictors are still not clarified. In this issue, we collected a registry-based longitudinal dataset that contained more than 7000 patients with fragility fractures treated surgically to detect potential predictors for clinical refracture. Methods: Based on the fact that machine learning algorithms are often used for the analysis of a large-scale dataset, we developed automatic prediction models and clarified the relevant features for patients with clinical refracture. Formats of input data containing perioperative clinical information were table data. Clinical refracture was documented as the primary outcome if the diagnosis of fracture was made at postoperative outpatient care. A decision-tree-based model, LightGBM, had moderate accuracy for the prediction in the test and the independent dataset, whereas the other models had poor accuracy or worse. Results: From a clinical perspective, rheumatoid arthritis (RA) and chronic kidney disease (CKD) were noted as the relevant features for patients with clinical refracture, both of which were associated with secondary osteoporosis. Conclusion: The decision-tree-based algorithm showed the precise prediction of clinical refracture, in which RA and CKD were detected as the potential predictors. Understanding these predictors may improve the management of patients with fragility fractures.

## 1. Introduction

Fragility fractures are associated with increased morbidity and an economic burden. In 1990, the number of fragility fractures was 1.7 million worldwide, which is estimated to rise to 6.3 million by 2050 [1]. This would surely induce an economic burden [2,3,4]. Scoring models for fragility fractures have been reported to precisely manage patients with risk factors [5]. Among these factors, patients with fragility fractures are at an increased risk of sustaining another fracture (refracture) [6]. To prevent refractures, fracture liaison services have been established, and risk factors for refractures have been investigated [5,7]. Recent studies have reported that patients with rheumatoid arthritis (RA), multiple fractures history, and old aged women are at an increased risk of fracture [6,8,9]. Although those predictors for refracture were crucial when managing patients with fragility fracture, no literature has reported on prediction models by machine learning algorithms for clinical refractures, to our knowledge.

Machine learning algorithms use large-scale clinical data and learn patterns to assess outcomes [10]. In the medical field, automatic prediction of cancer recurrence has been reported using artificial neural networks (ANNs) [10,11,12]. While ANNs are biologically based on studies of the nervous system, they aim for nonlinear regression of classification, and are not biologically realistic in their details [13]. The system comprises several layers of a computational unit (artificial neuron), in which connections of each unit are highly non-linear. Convolutional neural networks (CNNs), a class of ANNs, typically have specific layers: convolution layers and pooling layers. The two layers efficiently function as feature extractions for digital images [14]. In fact, CNNs are applied to make diagnoses based on images such as computed tomography, magnetic resonance imaging, radiography, ultrasound images, and pathological images [15,16,17,18,19,20].

In contrast, there is another algorithm, a decision tree model [21,22,23]. The main components of the model are nodes and branches, and an important step in building a model is splitting. As only input variables related to the target variables are used to split a parent node and make branches into child nodes, this algorithm is also based on non-parametric evaluation [24]. Previous studies have reported the effective use of the decision tree model for table data by extracting optimal features [25,26,27]. In practice, clinical data contain table data and/or images; thus, an appropriate model should be applied.

We previously investigated the refracture rates using the longitudinal cohort data of patients treated surgically for fragility fractures [28]. The number of registered patients was over 7000 and the format was table data, which allowed us to hypothesize that the prediction models would be built using decision-tree-based algorithms, ANN, or other popular models. The aims of this current study were (1) to compare the accuracy of machine learning algorithms that predict clinical refracture after fragility fracture treated surgically, and (2) to clarify the relevant features for patients with clinical refracture.

## 2. Materials and Methods

### 2.1. Study Design and Patients

This study was designed as a registry-based study approved by the local ethics committee of Hokkaido University Hospital (017-0448). Informed consent was obtained from all patients prior to inclusion. Figure 1 shows the flow chart of the data sets; the patients with non-vertebral fragility fractures treated surgically were registered. Among them, patients with two or more missing input data were excluded from the analysis. An independent dataset was created by randomly choosing one hospital. The remaining patients were also randomly subdivided into training (75%) and test (25%) datasets.

Input data were in table data format, which were collected as the following: sex; age; body mass index (BMI); primary fracture site (proximal part of the femur, proximal part of the humerus, and distal part of the radius); date of surgery; comorbidities including RA, diabetes mellitus, chronic kidney disease (CKD), and chronic obstructive pulmonary disease; presence of malignant tumor; warfarin use; glucocorticoid use; and pre- and post-operative treatments for osteoporosis (bisphosphonate, selective estrogen receptor modulator, teriparatide, and denosumab). We also investigated whether the durations of follow-ups were more than 24 months. In the case that the patients had symptoms such as pain at the postoperative outpatient follow-up, radiographs were taken and the diagnoses of clinical refracture were made.

### 2.2. Prepared Models

#### 2.2.1. Decision Tree Model

LightGBM was used as a decision-tree-based ensemble learning algorithm designed by Microsoft Research Asia [21]. Gradient boosting is a member of the ensemble learning paradigm. The learning procedure consecutively fits new models to provide a more accurate estimate. This is aimed to construct multiple weak learners to establish a more accurate and stronger model [24]. Although this ensemble part generates highly accurate models, there are several limitations, such as the unsatisfactorily long training time and scalability [29].

To solve some of these problems, LightGBM adopts a histogram algorithm and leaf-wise tree growth, which identifies the best leaf with the highest gain and only splits the best leaf, resulting in an asymmetrical tree [24]. This structure successfully decreased memory occupancy and improved accuracy compared with other variants [24,30]. In medical fields, this model has been applied successfully to assess each outcome [26,27].

#### 2.2.2. Feature Selection and Relative Importance

Feature selection was applied according to the implementation of LightGBM to detect the relevant features among the input data for clinical refracture. This is employed to remove redundant and irrelevant features to select the optimal feature subset [24,30]. Zhang stated that two kinds of importance types are contained in the LightGBM: one is “split”, and the other is “gain” [28]. “Split” contains the number of times the feature is used in a model, whereas “gain” reflects the total gains of splits which use the feature. Different from the “split,” the “gain” measures the actual decrease in node impurity. The feature rankings of gain-based importance can be obtained after LightGBM fitting [31,32], in which gain-based feature selection was adopted and relative importance was calculated.

#### 2.2.3. ANN: Artificial Neural Network Model

We implemented an ANN model that consists of dense layers, also known as fully connected layers, and activation layers with dropout layers. As the details of the ANN model are shown in Figure 2, each dropout layer was set to discard its value with a probability of 0.2. The sigmoid was adopted as the final activation layer. This model was modified from a previous study in the medical fields [10].

#### 2.2.4. SVM: Support Vector Machine Model

The SVM algorithm was originally proposed to construct a linear classifier in 1963 by Vapnik [33]. This algorithm aims to create a decision boundary between two classes that enables the prediction of labels from one or more feature vectors [34]. It enhances classification accuracy by plotting a multidimensional hyperplane that divides classes and increases the margins between classes [10,35]. In the medical field, it has been applied to colon cancer tissue classification using selected sequence data [36].

#### 2.2.5. Implementation Details

The experiments were performed on a computer comprising CPU^®^ Ryzen^TM^ 9 5950X @3.4 GHz, Advanced Micro Devices, Inc., Santa Clara, CA, USA; RAM 64 GB; and GPU NVIDIA^®^ GeForce RTX^TM^ 3090, NVIDIA Corporation, Santa Clara, CA, USA.

### 2.3. Statistical Analysis

Categorical variables were evaluated using the chi-square test, while continuous variables (age and BMI) were analyzed using the Mann−Whitney U test since they were not regarded as corresponding with the normal distribution using the Shapiro−Wilk test. Statistical analyses were conducted using a logistic regression model with JMP Pro version 14 (SAS Institute, Inc., Cary, NC, USA). The significance level was set at *p* < 0.05.

## 3. Results

### 3.1. Study Characteristics

Table 1 shows that the enrolled patients with a mean age of 77.2 years and the ratio of females was approximately 79.7%. As the main site of fragility fractures, 73.7% of the patients had fractures at the proximal femur. Post-operatively, 28.6% of the patients had postoperative treatments for osteoporosis (bisphosphonate, selective estrogen receptor modulator, teriparatide, and denosumab). In the cohort, the duration of follow-up for more than 24 months was 39.2%, and the incidence of clinical refracture was estimated to be 4.4%.

### 3.2. Comparison of the Models

There were no significant differences in patient demographic data, including the incidence of clinical refractures between the training and test sets, as shown in Table 2. LightGBM had an area under the curve (AUC) of 0.75 in the test dataset, as well as an AUC of 0.90 in the training dataset, whereas the ANN had an AUC of less than 0.60 in either set, as shown in Figure 3. Because the training was not successfully conducted by the SVM, this model could not describe the receiver operating curve.

### 3.3. Relevant Features by the LightGBM

LightGBM captured the relevant features of patients with clinical refracture during training. The higher relative importance of clinical refractures compared with no post-operative treatment, which tended to be associated with the incidence of refracture in our previous study [28], included CKD, RA, presence of malignant tumor, primary fracture site: proximal part of humerus, and warfarin use (Table 3). Glucocorticoid use scored 19.3 as relative importance, which was low compared with CKD, RA, or no post-operative treatments.

### 3.4. Assessments in the Independent Dataset

Figure 4 shows that LightGBM had an AUC of 0.74 in the independent dataset.

## 4. Discussion

We suggested a prediction model for clinical refracture after fragility fracture, which was performed using LightGBM, the decision-tree-based model. This model had an AUC of approximately 0.75 for prediction in the test dataset or independent dataset, whereas the other models had an AUC of less than 0.60 or worse. Considering that assessment models with an AUC of 0.70−0.90 are regarded as moderate [37], our model had moderate accuracy. In addition, RA and CKD were noted as the relative features of patients with clinical refracture.

When considering the clinical application of machine learning, appropriate algorithms should be selected to ensure an acceptable outcome. Two- or three-dimensional images have been accurately analyzed by CNN, a type of ANN [13,14,15,16,17]. Our report revealed, however, that the ANN had poor accuracy, whereas the decision-tree-based model showed satisfactory performance. The input data of our study were table data, and previous reports stated the effective use of the decision tree model for the table data [22]. Hence, our results are in accordance with this previous report.

Although machine learning algorithms assess the clinical outcome precisely, there is still a gap for clinical application; the black box covers the process until the output. In short, how do we understand the decisions suggested by machine learning [38]? To overcome this challenge, the concept of explainable artificial intelligence has been proposed. This field is concerned with the development of new methods that explain and interpret machine learning models [39]. In this study, for example, LightGBM was able to reveal the basis of the assessment according to feature importance. Similarly, feature importance and explainable artificial intelligence are linked [39].

Feature importance in our study showed the several potential predictors for clinical refractures, which can be divided into two groups. First, a group contained the items already detected by previous reports [9,28]. RA and no post-operative treatments for osteoporosis correspond to this. In another perspective, our model appeared reasonable since these items were detected as the top ranked features. In contrast, the other group contained unreported factors, such as CKD.

Control of phosphorus accumulation is crucial to prevent secondary osteoporosis [40]. Decreased glomerular filtration of phosphorus and hyperphosphatemia result in abnormal bone turnover and mineralization [41]. In fact, osteoporosis is more frequent in patients with CKD than those without CKD. To prevent fragility fracture in patients with CKD, adding vitamin D, reducing phosphate intake, and evaluating whether parathyroidectomy is essential is especially important [41]. In our study, CKD was detected as the top relevant feature of clinical refracture. As this is a well-known factor for secondary osteoporosis, further investigation is needed to verify the association with refracture.

RA was also detected as a relevant feature for clinical refracture, and RA is known to be a risk factor for refracture as well as fracture [7,9]. Osteoclasts are the main cell population responsible for bone loss in patients with RA [42]. For their differentiation, they require the intervention of macrophage colony-stimulating factor and the receptor activator of nuclear factor kappa B (RANK). As an osteoporosis treatment, denosumab inhibits osteoclast activity by targeting the RANK ligand [43]. Bisphosphonates also promote the apoptosis of osteoclasts, resulting in the suppression of bone turnover [44]. These mechanisms also support that no postoperative treatments for osteoporosis were detected as relevant features.

Previous studies showed the effectiveness of the medical intervention for refracture [45]; our previous report using part of the data in this study also showed the effectiveness of post-operative treatments for osteoporosis using the general statistics [28]. Thus, this study introduced the potential predictors, the relative importance of which scored equal to or greater than no post-operative treatments for the osteoporosis. Intriguingly, glucocorticoid use scored less than no post-operative treatments. Mono-variate analysis in our original data revealed that the patients with glucocorticoid use were significantly associated with RA, as well as clinical refractures (data not shown). This indicated the possibility of RA as a confounding factor. Further analyses should be conducted to clarify the roles of glucocorticoid according to duration or accumulation.

We devised an automatic prediction model for clinical refracture after a fragility fracture. The top relevant features of the refractures were involved in secondary osteoporosis. The precise prediction and understanding of the relevant features can lead to improved individual activity and avoidance of social burden.

This study had several limitations. First, the duration of follow-up was irregular among the enrolled patients. Second, this study was conducted only in Japan. Literature about fragility fracture in Japan also revealed female dominance among the population of fragility fracture [46]. Third, the enrolled patients were patients with non-vertebral fragility fractures who were treated with surgery alone. Fourth, cardiovascular diseases among the enrolled patients were not evaluated. Instead, we analyzed a related item—warfarin—which works against vitamin K, improving bone homeostasis and increasing bone mineral density [47,48]. Fifth, the long-term cohort (maximum: 10-year period) might have resulted in the fragility fracture and clinical refracture being less correlated.

## 5. Conclusions

We presented a prediction model with moderate accuracy for clinical refractures after fragility fractures using LightGBM, a decision-tree-based algorithm. Our report showed the effective use of this decision-tree-based algorithm for the table data format. From a clinical perspective, RA and CKD were noted as relevant features for patients with clinical refracture.

## Figures and Tables

**Figure 1 jcm-11-02021-f001:**
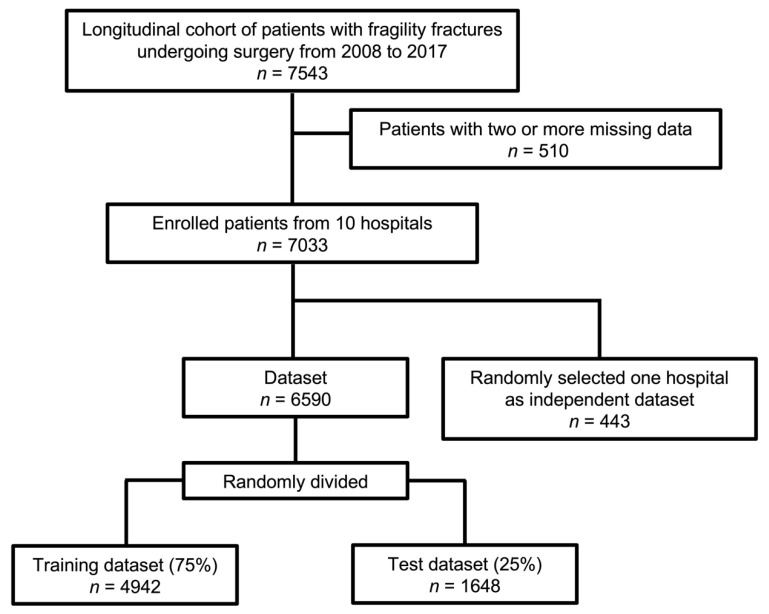
Study design.

**Figure 2 jcm-11-02021-f002:**
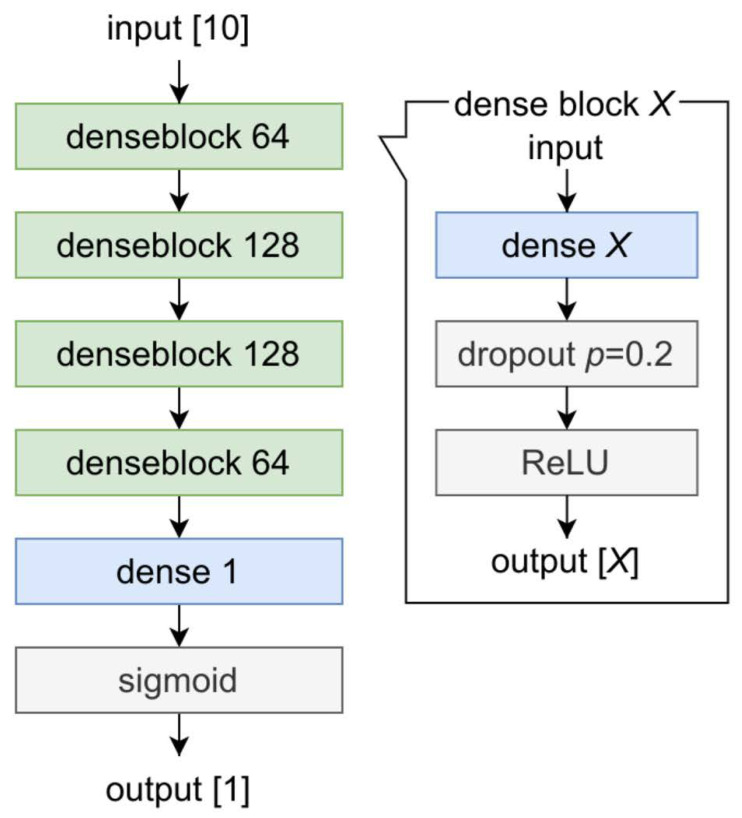
Structure of our artificial neural network (ANN) model. Denseblock is defined as a single block based on a combination of a dense layer, dropout layer and ReLU. The *X* corresponds to the dimension of the output tensor. ANN; artificial neural network: ReLU; rectified linear unit.

**Figure 3 jcm-11-02021-f003:**
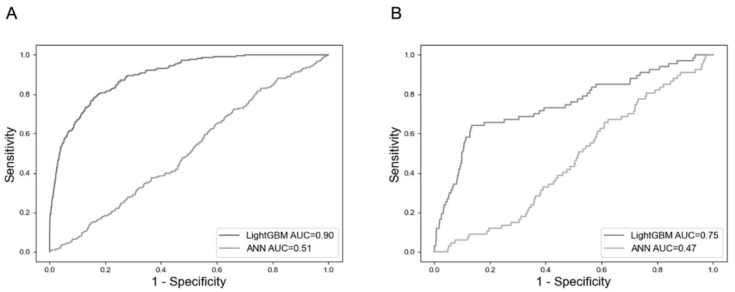
Comparison of receiver operating characteristics for the prediction of clinical refractures. (**A**) Evaluations in the training dataset. (**B**) Evaluations in the test dataset. ANN: artificial neural network; AUC: area under the curve.

**Figure 4 jcm-11-02021-f004:**
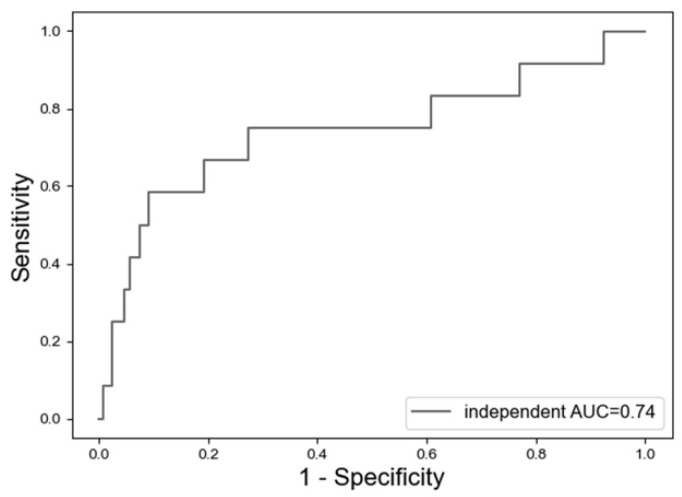
Receiver operating characteristics for the prediction of clinical refractures in the independent dataset using the LightGBM model. AUC: area under the curve.

**Table 1 jcm-11-02021-t001:** Demographic data of the enrolled patients.

Variables	
Sex (female)	79.7%
Age	77.2 ± 0.15
Body Mass Index	21.7 ± 0.06
Primary fracture site	
Proximal part of the femur	73.7%
Proximal part of the humerus	6.3%
Distal part of the radius	20.0%
Diabetes	19.1%
Chronic kidney disease	21.6%
Rheumatoid arthritis	2.6%
Chronic obstructive pulmonary disease	3.9%
Presence of malignant tumor	12.1%
Glucocorticoid use	2.8%
Warfarin use	5.2%
Pre-operative Ca or Vit. D	6.2%
Pre-operative treatments for osteoporosis	7.9%
Post-operative Ca or Vit. D	12.7%
Post-operative treatments for osteoporosis	28.6%
Follow-ups more than 24 months	39.2%

Data presented as mean (standard error of the mean). Ca: calcium; Vit. D: vitamin D3.

**Table 2 jcm-11-02021-t002:** Comparison of demographic data between the training and test sets.

Variables	Training Set	Test Set	*p*-Value
Sex (female)	79.6%	79.9%	0.584
Age	77.2 ± 0.18	77.2 ± 0.28	0.834
Body Mass Index	21.6 ± 0.06	21.8 ± 0.13	0.975
Primary fracture site			
Proximal part of the femur	73.6%	73.9%	0.837
Proximal part of the humerus	6.3%	6.1%	0.895
Distal part of the radius	19.9%	20.1%	0.758
Diabetes	19.6%	18.8%	0.459
Chronic kidney disease	21.6%	21.5%	0.915
Rheumatoid arthritis	2.7%	2.5%	0.785
Chronic obstructive pulmonary disease	3.8%	4.1%	0.588
Presence of malignant tumor	12.5%	11.1%	0.077
Glucocorticoid use	2.8%	2.7%	0.758
Warfarin use	5.0%	5.5%	0.333
Pre-operative Ca or Vit. D	6.0%	6.7%	0.262
Pre-operative treatments for osteoporosis	7.9%	8.0%	0.803
Post-operative Ca or Vit. D	12.9%	12.4%	0.63
Post-operative treatments for osteoporosis	28.7%	28.6%	0.92
Follow-ups more than 24 months	39.3%	39.0%	0.781

Data presented as mean (standard error of the mean). Ca: calcium; Vit. D: vitamin D3.

**Table 3 jcm-11-02021-t003:** Relative importance of the top six features among the categorical variables.

Feature Names	Relative Importance
Chronic kidney disease	52.1
Rheumatoid arthritis	31.4
Presence of malignant tumor	28.4
Primary fracture site: proximal part of humerus	27.8
Warfarin use	27.2
No post-operative treatments for osteoporosis	26.3

## Data Availability

The data presented in this study are available on request from the corresponding author.

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
