# Peer review of "Machine Learning Algorithms: Prediction and Feature Selection for Clinical Refracture after Surgically Treated Fragility Fracture"

_jcm, 2022, doi:10.3390/jcm11072021_

Round 1

Reviewer 1 Report

The work is carried out on a large number of people however there are very
important limitations and some corrections to be made.
1) Please describe ANN, CNN and decision tree better 2) Describe if there are any studies that they used the same model

3)
Line 66 specifies "fragility" fractures

4)
Describe LightGBM better

5)
Line 168: you should describe the type of treatment for osteoporosis

6)
Check the formatting of table 1

7)
Why has the use of corticosteroids not been evaluated?
They are held primarily responsible

8) Table 3:
Why did you choose the first six variables.
What makes them define relevant?

9)
Is there a numerical indicator? Is there a maximum or minimum limit?

10)
In the discussion line 239 we speak of "moderate" accuracy,
it explains the definition better

11)
There is no time interval for refracting. It may not be correlation
between fracture and refracture if it has been a long time

Author Response

Review 1:

The work is carried out on a large number of people however there are very important limitations and some corrections to be made.

1) Please describe ANN, CNN and decision tree better 2) Describe if there are any studies that they used the same model

Authors Response and Action: Thank you for pointing this out. Following the reviewer’s suggestion, we have added details of each machine learning algorithm. Additionally, the same model has also been reportedly used for the prediction of recurrence of breast cancer (Lous S-J et al. Cancers, 2021). We have revised the Introduction and Methods section (Page 2 Lines 48-56, Page 2 Lines 57-61 and Page 4 126-127) accordingly.

Page 2, Lines 48-56

While ANNs are biologically based on studies of the nervous system, they aim for nonlinear regression of classification, and are not biologically realistic in details [13]. The system comprises several layers of a computational unit (artificial neuron), in which connections of each unit are highly non-linear. Convolutional neural networks (CNNs), a class of ANNs, typically have specific layers: convolution layers and pooling layers. The two layers efficiently function as feature extractions for digital images [14]. In fact, CNNs are applied to make diagnoses based on images such as computed tomography, magnetic resonance imaging, radiography, ultrasound images, and pathological images [15-20].

Page 2, Lines 57-61

The main components of the model are nodes and branches, and an important step in building a model is splitting. As only input variables related to the target variables are used to split a parent node and make branches into child nodes, this algorithm is also based on non-parametric evaluation [24].

Page 4, Lines 126-127

This model was modified from a previous study in the medical fields [10].

3) Line 66 specifies "fragility" fractures

Authors Response and Action: Thank you for your careful review. We have revised the Methods section accordingly.

Page 3, Lines 77-78

Figure 1 shows the flow chart of the data sets; the patients with non-vertebral fragility fractures treated surgically were registered.

4) Describe LightGBM better

Authors Response and Action: Thank you for your valuable comment. As the reviewer suggested, we have added the details of LightGBM. We have revised the Methods section accordingly (Page 3 Line 98- Page 4 Line 108).

Page 3 Line 98- Page 4 Line 108

Gradient boosting is a member of the ensemble learning paradigm. The learning procedure consecutively fits new models to provide a more accurate estimate. This is aimed to construct multiple weak learners to establish a more accurate and stronger model [24]. Although this ensemble part generates highly accurate models, there are several limitations, such as the unsatisfactorily long training time and scalability [29].

To solve a part of these problems, LightGBM adopts a histogram algorithm and leaf-wise tree growth, which identifies the best leaf with the highest gain and only splits the best leaf, resulting in an asymmetrical tree [30]. This structure successfully decreases memory occupancy and improves accuracy compared with other variants [30,31]. In medical fields, this model has been applied successfully to assess each outcome [26,27]

5) Line 168: you should describe the type of treatment for osteoporosis

Authors Response and Action: Thank you for your careful review. The postoperative treatments for osteoporosis were bisphosphonate, selective estrogen receptor modulator, teriparatide, or denosumab. Therefore, we have revised the Results section accordingly. (Page 5 Lines 190-192)

Page 5, Lines 190-192

Post-operatively, 28.6% of the patients had postoperative treatments for osteoporosis (bisphosphonate, selective estrogen receptor modulator, teriparatide, and denosumab).

6) Check the formatting of table 1
Authors Response and Action: Thank you for your careful review. We have revised Table 1 accordingly.

Page 6, Line 201

Variables

Sex (female)

79.7%

Age

77.2±0.15

BMI

21.7±0.06

Primary fracture site

    Proximal part of the femur

73.7%

    Proximal part of the humerus

6.3%

    Distal part of the radius

20.0%

Diabetes

19.1%

Chronic kidney disease

21.6%

Rheumatoid arthritis

2.6%

Chronic obstructive pulmonary disease

3.9%

Presence of malignant tumor

12.1%

Glucocorticoid use

2.8%

Warfarin use

5.2%

Pre-operative Ca or Vit. D

6.2%

Pre-operative treatments for osteoporosis

7.9%

Post-operative Ca or Vit. D

12.7%

Post-operative treatments for osteoporosis

28.6%

Follow-ups more than 24 months

39.2%

Data presented as mean (Standard Error of the mean)

Ca: calcium; Vit. D: vitamin D3

7) Why has the use of corticosteroids not been evaluated?
They are held primarily responsible

Authors Response and Action: Thank you for your careful review. We agree with the reviewer’s point. We also evaluated the glucocorticoid use. Glucocorticoid use scored 19.3 as relative importance, which was low compared with CKD, RA, or no post-operative treatments. We have added the date of the glucocorticoid use in the Results section (Page 8, Lines 227-229).

Page 8, Lines 227-229

Glucocorticoid use scored 19.3 as relative importance, which was low compared with CKD, RA, or no post-operative treatments.

8) Table 3: Why did you choose the first six variables.
What makes them define relevant?
Authors Response and Action: Thank you for raising this important issue. As the reviewer has pointed out, we had some difficulties on how to evaluatethe relative importance because of the lack of general thresholds such as p-value in statistics. Our previous study reported that the post-operative osteoporosis treatment tended to be associated with the incidence of refracture (Shimodan et al. JBMM, 2020). According to that result, we mentioned the important risk factor for refracture in this manuscript. To mention the reason for choosing the first six variables, we have revised the Results section accordingly (Page 8 Lines 224-227).

Page 8 Lines 224-227

The higher relative importance of clinical refractures compared with no post-operative treatment, which tended to be associated with the incidence of refracture in our previous study [28], were CKD, RA, presence of malignant tumor, primary fracture site: proximal part of humerus, and warfarin use (Table 3).

9) Is there a numerical indicator? Is there a maximum or minimum limit?
Authors Response and Action: Thank you for your careful review.

The valuesof relative importance were extracted as a whole when LightGBM evaluated the training dataset with the specific seed in which the model scored 0.90 area under curve. Therefore, there were no maximum or minimum limits. We have corrected the significant digits and the format in Table 3.

Page 8, Line 226

                    Feature names

Relative importance

Chronic kidney disease

52.1

Rheumatoid arthritis

31.4

Presence of malignant tumor

28.4

Primary fracture site: proximal part of humerus

27.8

Warfarin use

27.2

No post-operative treatments for osteoporosis

26.3

10) In the discussion line 239 we speak of "moderate" accuracy,
it explains the definition better

Authors Response and Action: Thank you for your careful review. Because Akobeng et al. reported that an assessment model with an AUC of 0.70−0.90 is regarded as moderate [38], we think that this model was moderate accuracy. To avoid confusion, we have revised the Results and Discussion sections.  

Page 6, Lines 209-211

LightGBM had an area under the curve (AUC) of 0.75 in the test dataset as well as an AUC of 0.90 in the training dataset, whereas the ANN had an AUC of less than 0.60 AUC in either set, as shown in Figure 3.

Page 8, Line 234

Figure 4 shows that LightGBM had an AUC of 0.74 in the independent dataset.

Page 8, Lines 256-260

This model had an AUC of approximately 0.75 for prediction in the test dataset or independent dataset, whereas the other models had an AUC of less than 0.60 or worse. Considering that assessment models with an AUC of 0.70−0.90 are regarded as moderate [38], our model had moderate accuracy.

11) There is no time interval for refracting. It may not be correlation between fracture and refracture if it has been a long time

Authors Response and Action: Thank you for your thoughtful comment. As the reviewer pointed out, there could be less correlation. However, it might be difficult to distinguish the correlation. Therefore, we have added this possibility as a limitation of this study. (Page 10, Line 322-323)

Page 10, Lines 321-322

Fifth, the long-term cohort (maximum: 10 years period) might have let the fragility fracture and clinical refracture to be less correlated.

Reviewer 2 Report

Abstract

- The abstract conclusion is too verbose. social burden? Did this article discuss social burden or economics? The conclusion part needs to be described in more detail. Rather, it is better to match the conclusion section of the main text.

Introduction

- Well-organized, concise.

Materials and methods

-Line72-81: The proportion of patients with cardiovascular disease in enrolled patients must also be significantly higher. Were there any comorbidities that excluded cardiovascular disease?

Why include warfarin alone among so many other anticoagulants, including NOACs? Was this the most representative of the anticoagulants?

-Line 154-155: In order to use the product name, please also indicate "company name, region, country".

-Line 157-161: Was the normal distribution of the data been evaluated? Shouldn't that be a premise for all statistical analysis?

Results

-Line 167-168: even though the rate of treatment for postoperative osteoporsis (28.6%) was the highest, it seems that more discussion is needed about what is not included in the relevant features. It is necessary to check whether there is a problem with the machine learning process.

-Table 1: What was the difference between taking Ca or Vit D and treatment for osteoporosis? Was there any particular reason you shared this?

-Line 210: The most relevant features for patients with clinical ‘refracture’

Discussion

-Line 256-258: CKD, a major relevant feature of clinical refracture, should also be discussed in more detail in the discussion section.

-Line 263-266: the limitation section was too short. It seems necessary to supplement or defend the weaknesses suggested by the authors. Also missing about female predominance.

Conclusion

-Reasonable and concise.

Author Response

Review 2

Abstract

- The abstract conclusion is too verbose. social burden? Did this article discuss social burden or economics? The conclusion part needs to be described in more detail. Rather, it is better to match the conclusion section of the main text.

Authors Response and Action: Thank you for the careful review. We agree with the reviewer’s suggestion. We have revised the Abstract to match the conclusion section of the main text.

Page 1, Lines 13-15

The number of patients with fragility fracture has been increasing. Although the increasing number of patients with fragility fracture increased the rate of fracture (refracture), the causes of refracture are multifactorial, and its predictors are still not clarified.

Page 1, Lines 26-29

The decision-tree-based algorithm showed the precise prediction of clinical refracture, in which RA and CKD were detected as the potential predictors. Understanding these predictors may improve the management of patients with fragility fractures.

Introduction

- Well-organized, concise.

Authors Response and Action: Thank you for your favorable comment. 

Materials and methods

-Line72-81: The proportion of patients with cardiovascular disease in enrolled patients must also be significantly higher. Were there any comorbidities that excluded cardiovascular disease?

Why include warfarin alone among so many other anticoagulants, including NOACs? Was this the most representative of the anticoagulants?

Authors Response and Action: Thank you for your careful review. As the reviewer pointed out, we agree that there is an association of osteoporotic fracture with severe cardiovascular diseases, such as myocardial infraction and stroke. To treat them, warfarin has been considered, and warfarin itself functions against Vitamin K. As Vitamin K improves bone quality, we focused on the use of warfarin as a related factor to cardiovascular disease. We have added the fact that we did not evaluate cardiovascular diseases in the limitations section.

Page 10, Lines 318-321

Fourth, cardiovascular diseases among the enrolled patients were not evaluated. Instead, we analyzed a related item: warfarin; it works against vitamin K, which improves bone homeostasis and increases bone mineral density [49,50].

-Line 154-155: In order to use the product name, please also indicate "company name, region, country".

Authors Response and Action: Thank you for your careful review. We have added the details of the product.

Page 5, Lines 175-177

The experiments were performed on a computer comprising CPU® RyzenTM 9 5950X @3.4 GHz, Advanced Micro Devices, Inc, California, US; RAM 64 GB; and GPU NVIDIA® GeForce RTXTM 3090, NVIDIA Corporation, California, US.

-Line 157-161: Was the normal distribution of the data been evaluated? Shouldn't that be a premise for all statistical analysis?

Authors Response: Thank you for your careful review. We re-performed the statistical analysis and revised the statistical method.

Page 5, Lines 180-184

Categorical variables were evaluated using the chi-square test, while continuous variables (age and BMI) were analyzed using the Mann−Whitney U test since they were not regarded as corresponding with the normal distribution using the Shapiro−Wilk Test. Statistical analyses were conducted using a logistic regression model with JMP Pro ver-sion 14 (SAS Institute, Inc., Cary, NC, USA). The significance level was set at p < 0.05.

Results

-Line 167-168: even though the rate of treatment for postoperative osteoporsis (28.6%) was the highest, it seems that more discussion is needed about what is not included in the relevant features.

Authors Response and Action: Thank you for your important comments. As reviewer 1 has suggested, the use of glucocorticoids did not have high feature importance contrary to our expectation. We have added the related comments in the Results (Page 8 Lines 227-229) and Discussion (Page 9, Lines 304-309) sections about post-operative treatments for osteoporosis and glucocorticoid.

Page 8, Lines 227-229

Glucocorticoid use scored 19.3 as relative importance, which was low compared with CKD, RA, or no post-operative treatments.

Page 9, Lines 304-309

Intriguingly, glucocorticoid use scored less than no post-operative treatments. Mono-variate analysis in our original data revealed that the patients with glucocorticoid use were significantly associated with RA as well as clinical refractures (data not shown). This indicated the possibility of RA as a confounding factor. Further analyses should be conducted to clarify the roles of glucocorticoid according to duration or accumulation.

It is necessary to check whether there is a problem with the machine learning process.

Authors Response and Action: Thank you for your important suggestion. We are frequently faced with difficulties when checking whether the machine learning process functions without problems. To solve a part of this issue, the concept of explainable machine learning has been argued. As this study aimed to clarify the basis of the outcome, feature importance is generally said to be associated with this concept. In our study, we showed the potential predictor, which contained no postoperative treatments for osteoporosis and rheumatoid arthritis. They both were already reported as risk factors for clinical fractures. We consider these facts supplemented the right process in our model. We have also added comments about the process of machine learning in the Discussion section (Page 9, Lines 263-276).

Page 9, Lines 269-276

Although machine learning algorithms assess the clinical outcome precisely, there is still a gap for clinical application; the black box covers the process until the output. In short, how do we understand the decisions suggested by machine learning? [39]. To overcome this challenge, the concept of explainable artificial intelligence has been proposed. This field is concerned with the development of new methods that explain and interpret machine learning models [40]. In this study for example, LightGBM was able to reveal the basis of the assessment according to feature importance. Similarly, feature importance and explainable artificial intelligence are linked [40].

Lines 277-281

Feature importance in our study showed the several potential predictors for clinical refractures, which can be divided into two groups. First, a group contained the items already detected by previous reports[9,28]. RA and no post-operative treatments for osteoporosis correspond to this. In another perspective, our model appeared reasonable since these items were detected as the top ranked features.

-Table 1: What was the difference between taking Ca or Vit D and treatment for osteoporosis? Was there any particular reason you shared this?

Authors Response and Action: Thank you for your careful review. As you rightly pointed out, both Vitamin D and calcium improve bone quality. But those are often argued separately in the research area, and in clinical practice, bisphosphonates or denosumab are frequently prescribed with Vitamin D or calcium. Additionally calcium and Vitamin D are approved as a supplement by our country. This is why we regarded the medications separately.

-Line 210: The most relevant features for patients with clinical ‘refracture’

Authors Response and Action: Thank you for your review. We have revised this paragraph (Page 8 Lines 224-227).  

Page 8, Lines 224-227

The higher relative importance of clinical refractures compared with no post-operative treatment, which tended to be associated with the incidence of refracture in our previous study [28], were CKD, RA, presence of malignant tumor, primary fracture site: proximal part of humerus, and warfarin use (Table 3).

Discussion

-Line 256-258: CKD, a major relevant feature of clinical refracture, should also be discussed in more detail in the discussion section.

Authors Response and Action: Thank you for the suggestion. As you have mentioned, CKD is an important factor for secondary osteoporosis. As more details should have been described, we have added the related comments in the Discussion section.

Page 9, Lines 281-290

In contrast, the other group contained unreported factors, such as CKD.

Control of phosphorus accumulation is crucial to prevent secondary osteoporosis [41]. Decreased glomerular filtration of phosphorus and hyperphosphatemia result in abnormal bone turnover and mineralization [42]. In fact, osteoporosis is more frequent in patients with CKD than those without CKD. To prevent fragility fracture in patients with CKD, adding vitamin D, reducing phosphate intake, and evaluating whether parathyroidectomy is required is especially important [42]. In our study, CKD was detected as the top relevant feature of clinical refracture. As this is a well-known factor for secondary osteoporosis, further investigation is needed to verify the association with refracture.

-Line 263-266: the limitation section was too short. It seems necessary to supplement or defend the weaknesses suggested by the authors. Also missing about female predominance.

 Authors Response and Actions: Thank you for your suggestion. As you have mentioned, other limitations were also suggested. We have added the limitations which comprised female predominance (Page 9 Lines 314- Page 10 Line 322).

Page 9 Line 314- Page 10 Line 322

This study had several limitations. First, the duration of follow-up was irregular among the enrolled patients. Second, this study was conducted only in Japan. Literature about fragility fracture in Japan also revealed female dominance among the population of fragility fracture [48]. Third, the enrolled patients were patients with non-vertebral fragility fractures who were treated with surgery alone. Fourth, cardio-vascular diseases among the enrolled patients were not evaluated. Instead, we analyzed a related item: warfarin; it works against vitamin K, which improves bone homeostasis and increases bone mineral density [49,50]. Fifth, the long-term cohort (maximum: 10 years period) might have let the fragility fracture and clinical refracture become less correlated.

Conclusion

-Reasonable and concise.

Authors Response and Action: Thank you for your favorable comment.

Reviewer 3 Report

The revision is adequate.

Please ensure the figure at page 4 is properly attached.

Round 2

Reviewer 1 Report

The corrections were done very well. You could improve the introduction by helping yourself with the following
articles:
1) Does the combination of therapeutic exercise and supplementation with sucrosomal magnesium improve posture and balance and prevent the risk of new falls?
2) Is there a relationship between mild to moderate back pain and fragility fractures? Original investigation